# The Efficiency of Financing Environmental Protection Measures in the Context of Ukraine's Future Membership in the EU

**Oleksandr Labenko [1,2,\*], Andrjus Sadauskis [3] and Valeriia Lymar [4]**

[1] Faculty of Economics, University of Applied Sciences, 08248 Vilnius, Lithuania
[2] Faculty of Economics, National University of Life and Environmental Sciences of Ukraine, 03041 Kyiv, Ukraine
[3] Faculty of Public Governance and Business, Mykolas Romeris University, 08303 Vilnius, Lithuania; a.sadauskis@viko.lt
[4] International Relations and Foreign Policy Department, Vasyl' Stus Donetsk National University, 21021 Vinnytsia, Ukraine; limval555555@gmail.com
[\*] Correspondence: labenko@nubip.edu.ua

**Abstract:** In recent decades, humanity has had a significant negative impact on the environment. This problem can be solved only by establishing a rational environmental management policy and ensuring an effective financial policy in the context of balancing emissions and expenditures on environmental protection measures. The purpose of this article is to analyze the efficiency of financing environmental protection measures by determining the dependence of pollutant emissions on environmental protection expenditures in Ukraine and the European Union. The following methods were used for the study: analysis, synthesis, generalization, comparison, specification, and statistical and graphical methods. The statistical information was systematized on the basis of open data from the Open Budget web portal, the State Statistics Service, and Eurostat. This study identified the dynamics of revenues from environmental taxes in Ukraine; taxes on air emissions prevail. Most of the expenditures are made from the state budget. According to the functional classification, environmental expenditures are mainly aimed at preventing and eliminating environmental pollution. The dynamics of capital investments in environmental protection were also studied. To assess the effectiveness of the state policy in the field of environmental protection, we analyzed the dependence of pollutant emissions in Ukraine, Poland, and Romania on the amount of environmental expenditures and investments in this area and revenues from environmental tax. In Ukraine, the amount of pollutants released into the atmosphere depends mostly on investments in this area; in Poland—on revenues from environmental taxes; and in Romania—on expenditures on environmental protection. It has been established that the obtained models are adequate and can be used to build future forecasts of pollutant emissions. Directions for the development of financial and environmental policy are proposed. Post-war restoration of the environmental situation should be carried out on the basis of sustainable development, focusing on the European Green Deal A triple-task approach should be implemented, including environmental restoration, the minimization of negative climate change and balanced use of resources, and the expansion of powers of the relevant ministry with a focus on the strategic goals of the state policy. It is necessary to develop methodological recommendations according to international standards to assess the real state of the environment.

**Keywords:** environmental protection expenditures; environmental taxes; pollutant emissions; investments

## 1. Introduction

Globalization is having a significant impact on people's lifestyles [1]. It is transforming socio-economic forms of development within the existing natural resource and environmental conditions [2]. Globalization increases communication, accelerates access to technology and innovation, and has ushered in an era of economic prosperity [1]. On the other hand,

all of this has a negative impact on the environment and is a significant problem of the 21st century [3], provoking climate change, droughts, fires, etc. [4]. Environmental protection, the rational use of resources, and ensuring the ecological protection of human life are prerequisites for sustainable economic and social development of any country [5]. Ensuring ecological safety, namely the protection and restoration of the environment, is a priority task of the state and society [6]. One of the important areas of this sector is the alignment of financial flows with sustainable practices to achieve long-term climate and development goals [7]. The main directions of modern socio-environmental policy are the greening of social production and ensuring the environmental safety of the population and natural ecosystems [8]. The solution of environmental problems largely depends on the efficiency of the financial support system, the established composition and volume of funding sources, and the identified priority areas of their use, which requires a scientifically based analysis [9].

The object of this study is Ukraine, a developing country that is currently in the midst of military operations. It requires significant transformational changes in environmental policy and development strategy to address the environmental challenges it is currently facing. As Ukraine seeks to become a member of the EU and develop a course toward achieving the Sustainable Development Goals, it is important to assess the environmental component in comparison with other member states. We chose Poland and Romania as the countries bordering Ukraine and most closely related to environmental development.

In Ukraine, the main sources of public funding for environmental protection are currently the state and local budgets [1]. In the structure of environmental financing, the key place is occupied by environmental taxes and fees, the proceeds from which are used to protect the environment, minimize the negative impact of economic activity, and manage the rational use of natural resources [10]. At the same time, in the current system of regulation in the field of environmental management and environmental taxation in Ukraine, the existing levels of payments and fees are not able to ensure sustainable development in the accumulation of financial resources and the targeted allocation of funds for environmental activities [11]. Therefore, the implementation of successful foreign experience in this area to the Ukrainian economic space can yield positive results at the initial stages of reform [12].

Studies on environmental expenditures have been carried out by scholars such as Caglar A. E., Yavuz E. [4], Glukhova V., Kravchenko K. [5], Pirgaip B., Bayrakdar S., Kaya M. V. [7], Bukalo N. [8], Yaroshevych N. and Yakymiv A. [9], Karlin M., Prots N., Prots V. [10], Cherenkevych O. [11], Samko O. [12], and others. The trends of the green economy have been studied by Zhang L., Xu M., Chen H., Li Y., and Chen S. [1]. Despite the numerous studies, the issue of financing the environmental sector, including in Ukraine, remains quite relevant, especially now that society is facing global threats of climate change, ecosystem degradation, natural disasters, and human-made impacts.

The purpose of this article is to analyze the efficiency of financing environmental protection measures by determining the dependence of pollutant emissions on environmental protection expenditures in Ukraine and the European Union. The contribution of this study is that the proposed models can be used to build future forecasts of pollutant emissions and to develop state policy in the field of regulating environmental protection financing.

This paper consists of an introduction, materials and methods, results, a discussion, and conclusions. The Section 3 presents an analysis of the revenues from environmental taxes, environmental expenditures by budget and functional classification, and capital investment in environmental protection.

## 2. Materials and Methods

The main research period is 2015–2021. This time period allowed us to form an objective vision of the situation, since the statistics contain information on the temporarily occupied territories until 2014, and in 2022, due to the beginning of the full-scale invasion, environmental protection expenditures decreased significantly due to objective factors and

there is no information on the territories temporarily occupied after 24 February 2022. To understand the scale of the problem that currently exists in Ukraine, we have additionally displayed some indicators for 2022 and calculated projected indicators for 2023.

To form a statistical sample for the correlation and regression model, the time interval of 2015–2020 was chosen. Time periods before 2015 were not analyzed, as they contain somewhat outdated information and are incomparable due to the beginning of Russian aggression. Information about 2021 is not available in the statistics, and the figures for 2022 are incomparable due to the beginning of Russia's full-scale invasion, which has affected environmental policy as well. Therefore, it is illogical to use such data to determine certain patterns under optimal sustainable conditions, and the model is based on data for 2015–2020.

The statistical information was systematized on the basis of open data from the Open Budget web portal, the State Statistics Service, and Eurostat.

The following methods were used for this study: analysis, synthesis, generalization, comparison, specification, and statistical and graphical methods.

Using correlation and regression analyses, the influence of factors on the emissions of major pollutants into the atmosphere was assessed. Emissions of the main pollutants into the atmosphere are defined as a resultant feature. Expenditures, investments, and environmental tax revenues were chosen as the factor attributes. At the first stage, correlation matrices were constructed using the Microsoft Excel 2013 data block "Data Analysis" (the "Correlation" tool) to determine the correlation between the dependent and independent variables. The correlations range from −1 to +1, where −1 is a perfect negative correlation, 0 is no correlation, and +1 is a perfect positive correlation. Based on the results of the correlation analysis, the data with the highest level of correlation were selected for regression analysis.

Then, based on the formed sample, a regression analysis was conducted, which shows the contribution of the independent variable to the variation in the dependent variable under study. This analysis was conducted using the Microsoft Excel data block "Data Analysis" (the "Regression" tool). The regression analysis allowed us to estimate the size and direction of the relationship. As a result of the analysis, linear regressions were generated, which generally look like this:

$$y = b_0 + b_1 x_1, \tag{1}$$

y—dependent variable;
x—independent variable;
$b_0$, $b_1$—regression coefficients.
For Ukraine, the linear regression equation is as follows:

$$y = 4984.5 + (-6.8)x_1 \tag{2}$$

For Poland, the linear regression equation is as follows:

$$y = 450{,}013.0 + (-160.2)x_1 \tag{3}$$

For Romania, the linear regression equation is as follows:

$$y = 116{,}392.9 + (-11.5)x_1 \tag{4}$$

## 3. Results

Tax policy is a part of the government's toolkit to address environmental issues, including climate change. Environmental taxation can help reduce environmentally harmful behavior while generating revenue at all levels of government [13]. The national legislation emphasizes that the main purpose of establishing an environmental tax is to increase incentives for the rational use of natural resources [14].

In Ukraine, the environmental taxes include:

- An environmental tax levied on the emissions of pollutants into the atmosphere by stationary sources of pollution (except for emissions of carbon dioxide into the atmosphere);
- Revenues from pollutant disposals directly into water bodies;
- Revenues from waste disposal in specially designated places or facilities, except for the disposal of certain types of waste as secondary raw materials;
- An environmental tax levied on the generation of radioactive waste (including already accumulated waste) and/or the temporary storage of radioactive waste by its producers beyond the period established by the special conditions of licenses;
- An environmental tax levied on carbon dioxide emissions from stationary sources of pollution (Figure 1).

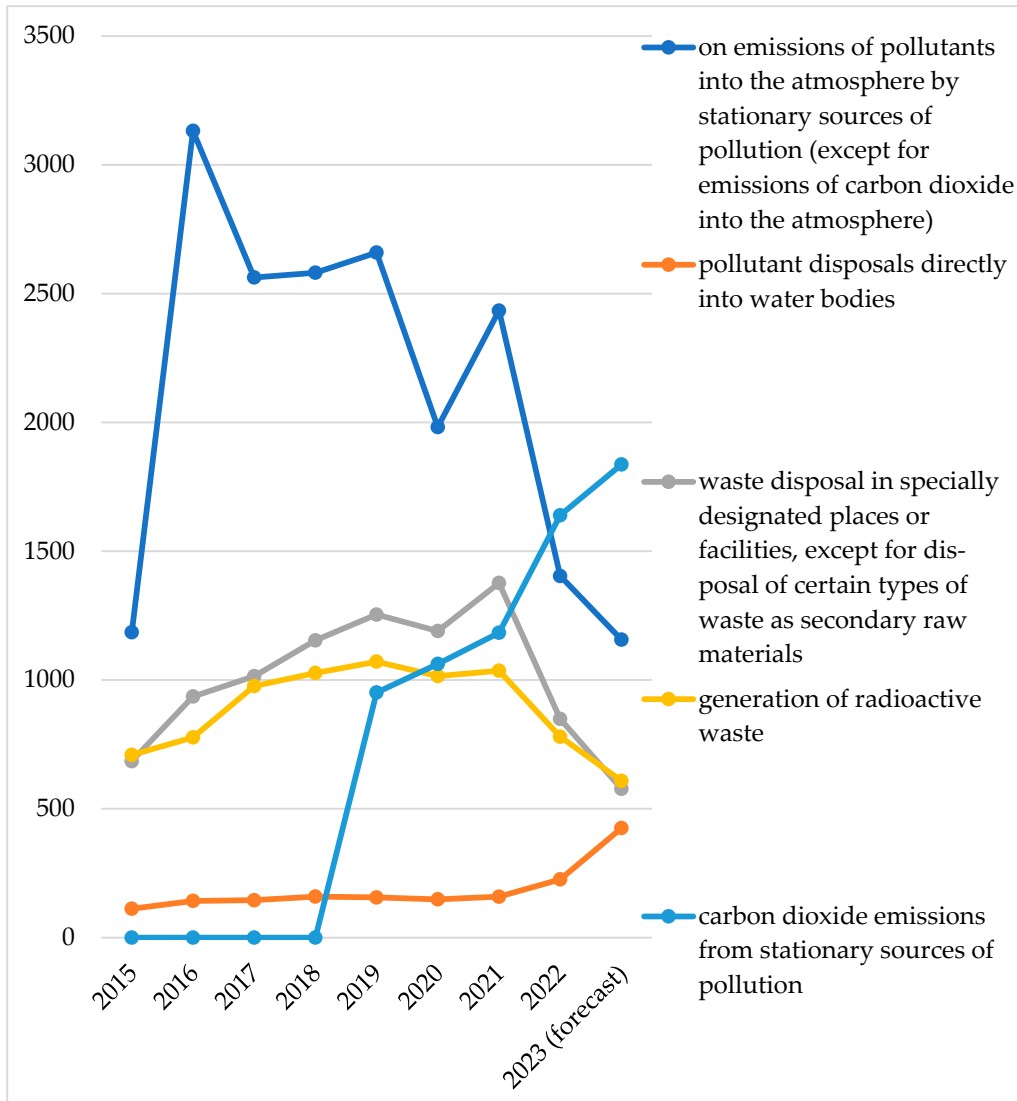

**Figure 1.** Revenues from environmental taxes (according to the consolidated budget of Ukraine, millions UAH). Source: based on data from [15].

As can be seen in Figure 1, taxes on air emissions account for the largest share of the total amount of environmental taxes. In total, in 2015–2023, including the taxes on carbon dioxide emissions, they amounted to 44.1%, 62.8%, 54.5%, 52.5%, 43.7%, 37.6%, 39.3%, 28.7%, and 25.1%, respectively. The share of taxes on emissions into water bodies was 5% in 2015–2022 and is projected to reach 9.2% in 2023. The maximum share of revenues from the disposal of waste in specially designated places during 2015–2022 was 25.4% in 2015, and the (projected) minimum share was 12.5% in 2023.

The share of the environmental tax levied for the generation and/or storage of radioactive waste in the total amount of environmental taxes in 2015–2021 averaged 19.1%. From 2020 to 2023, the share is projected to decrease by 5.6%.

The share of the environmental tax levied on carbon dioxide emissions from stationary sources of pollution increased to 24.3% in 2019–2023.

The share of the environmental tax on carbon dioxide emissions increased from 15.6% to 39.9% in 2019–2023.

Revenues from the environmental tax on air emissions, excluding carbon dioxide, increased significantly in 2016 compared to 2015 (+164.2%), and then the amount of funds decreased by 18.2% and remained almost unchanged in 2018–2019. In 2020, compared to 2019, there was a decline of 25.5%, followed by an increase of 22.8% and a further significant chain reduction of 42.3%. In 2023, the decrease in revenues is projected to be 17.6%.

In 2016–2018 and in 2021, there were increases in the revenues from discharges into water bodies of 27.3%, 1.8%, 9.8%, and 7.1% compared to the previous years, respectively. Decreases in the revenues occurred in 2019 and 2020 (−2.1% and −4.9%, respectively). In 2022, there was a sharp increase compared to 2021 (+42.4%). Growth is also predicted for 2023 (+88.1%).

In 2015–2019, the total increase in revenues from waste disposal in designated areas was 67.5%. During 2019–2022, their value changed in waves: in 2019 and 2021, there were increases (+8.7% and 15.7%, respectively), and in 2020 and 2022, decreases (−5.1% and −38.4%, respectively). The decrease in these revenues in 2023 may amount to 32.0%.

The maximum decrease in revenues from the environmental tax levied on the generation and/or temporary storage of waste occurred in 2022 compared to 2021 (−24.7%). The maximum increase in revenues from the environmental tax levied on the generation and/or temporary storage of waste occurred in 2017 compared to 2016 (+25.6%).

Revenues from the environmental tax on carbon dioxide emissions are characterized by stable growth (+11.7% in 2020, +11.4% in 2021, +38.5% in 2022, and +12.0% in 2023 (forecast), compared to the previous periods).

While analyzing revenues from environmental taxes, it is important to examine the expenditure component as an instrument of a coherent environmental and fiscal policy. The expenditures on environmental protection include all expenditures on preventing, minimizing, or eliminating negative effects. Table 1 shows the environmental expenditures by type of budget and in total.

**Table 1.** Environmental protection expenditures in Ukraine.

| Year | Local Budgets, Millions UAH | | State Budget, Millions UAH | | Consolidated Budget, Millions UAH | |
|---|---|---|---|---|---|---|
| | Total, Millions UAH | Per 1 Person, UAH | Total, Millions UAH | Per 1 Person, UAH | Total, Millions UAH | Per 1 Person, UAH |
| 2015 | 1477 | 34 | 4053 | 95 | 5530 | 129 |
| 2016 | 1484 | 35 | 4772 | 112 | 6255 | 147 |
| 2017 | 2609 | 61 | 4740 | 112 | 7349 | 173 |
| 2018 | 3001 | 71 | 5241 | 124 | 8242 | 195 |
| 2019 | 3414 | 81 | 6316 | 151 | 9730 | 232 |
| 2020 | 3777 | 90 | 7433 | 178 | 11,211 | 268 |
| 2021 | 3266 | 78 | 9299 | 223 | 12,565 | 301 |
| 2022 | 513 | 14 | 4714 | 131 | 5227 | 145 |
| 2023 (prediction) | 760 | | 4437 | | 5195 | |

Source: systematized according to [16].

In 2015–2021, the local environmental expenditures increased by UAH 1789 million (121.1%), while the state budget expenditures increased by UAH 5246 million (129.4%). According to the consolidated budget, the expenditures increased by UAH 7035 million, which is 127.2%. For a more objective assessment, we analyzed these expenditures per

capita, as it is important not only to increase expenditures in general, but to do so in proportion to the population. In 2015–2021, the expenditures per capita increased by UAH 172 (+133.3%), including UAH 128 (+134.7%) from the state budget and UAH 44.0 (+129.4%) from local budgets. That is, the expenditures per capita are growing at a higher rate than the total expenditures.

Let us take a closer look at 2022. The total expenditures on environmental protection decreased by UAH 7338 million (−58.4%) from UAH 12,565 million to UAH 5227 million, including UAH 4585 million (−49.3%) from the state budget and UAH 2753 million (−84.3%) from local budgets. The expenditures per capita decreased by UAH 156 (−51.8%); the state budget expenditures—by UAH 92 (−41.3%); and the local budgets—by UAH 64 (−82.1%).

Since the budget deficit is usually measured as a percentage of GDP, we set the following. The expenditures as a percentage of GDP according to the local budget in 2015–2021 amounted to 0.1%, and in 2022, this figure decreased. According to the state budget, the value averaged 0.2%, except for 2018 and 2022, when the figure was 0.1%. According to the consolidated budget, this value was 0.3% in 2015–2016 and 2020, and 0.2% in 2017–2019 and 2021; in 2022, the share decreased to 0.1%.

Next, let us analyze the environmental expenditures in more detail by the functional classification of expenditures (Figure 2 and Table 2).

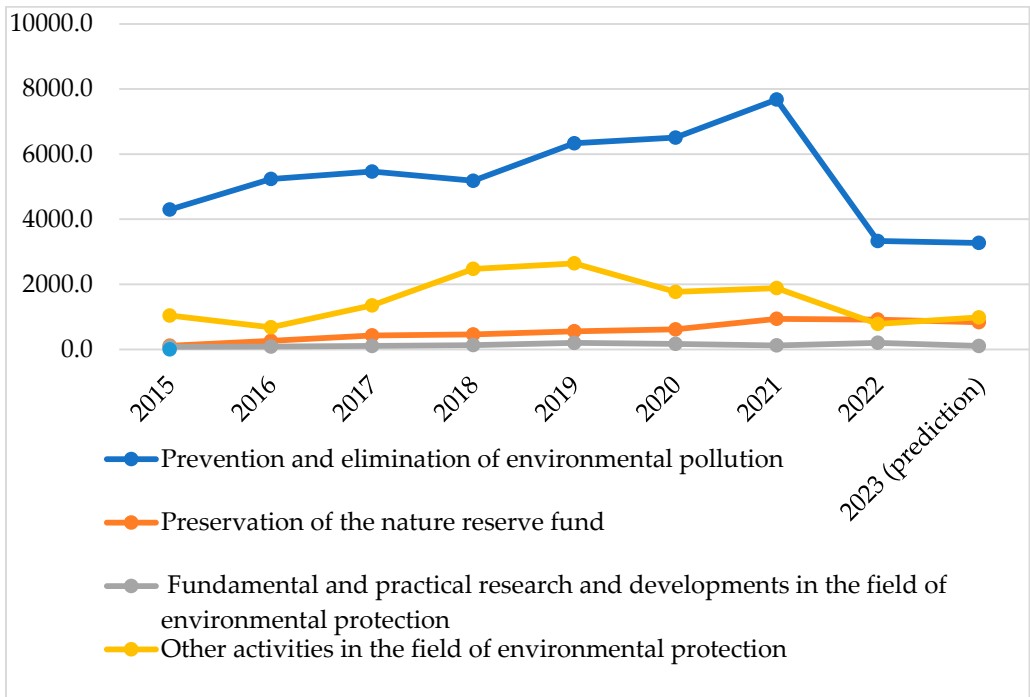

**Figure 2.** Ukraine's environmental protection expenditures by functional classification, millions UAH. Source: based on data from [17].

In 2015–2022, the largest share of environmental protection expenditures was allocated to prevent and eliminate environmental pollution. During 2015–2018, according to the state budget, their share decreased from 82.2% to 69.8%, and during 2018–2022, it increased from 69.8% to 81.6% (+11.8%); furthermore, there was a 14.9% decrease in expenditures (from 81.6% to 66.7%). According to the local budget indicators for 2018–2022, these expenditures decreased by 14.9%. During 2018–2021, in the consolidated budget, expenditures on the prevention and elimination of environmental pollution increased by 9.5%, and in 2022, decreased by 14.0% compared to 2021. The projected calculation for 2023 reflects an increase of 2.9%. The bulk of expenditures were made at the expense of the state budget. During 2018–2021, there was an increase in the consolidated budget of 48.2%, including an increase

in the state budget of 80.7% and a decrease in the local budget of 30.0%. In 2022, revenues to local budgets decreased by 82.6%, the state budget by 52.5%, and the consolidated budget by 56.7%.

**Table 2.** Expenditures on environmental protection in Ukraine by functional classification in terms of budgets, millions UAH.

| Year | Prevention and Elimination of Environmental Pollution | | Preservation of the Nature Reserve Fund | | Fundamental and Practical Research and Developments in the Field of Environmental Protection | | Other Activities in the Field of Environmental Protection | |
|---|---|---|---|---|---|---|---|---|
| | State | Local | State | Local | State | Local | State | Local |
| 2015 | 3331.5 | 963,6 | 53.9 | 59.5 | 81.3 | | 586.2 | 453.6 |
| 2016 | 4054.8 | 1179.8 | 209.6 | 49.7 | 84.8 | | 422.5 | 254.4 |
| 2017 | 3651.1 | 1813.6 | 361.6 | 66.6 | 104.3 | | 622.8 | 729.0 |
| 2018 | 3660.8 | 1519.4 | 420.0 | 39.1 | 130.4 | | 1029.9 | 1442.2 |
| 2019 | 4774.1 | 1559.3 | 501.6 | 52.3 | 197.4 | | 842.9 | 1801.7 |
| 2020 | 5416.0 | 1091.9 | 549.7 | 64.7 | 167.5 | | 503.5 | 1263.1 |
| 2021 | 6616 | 1062.9 | 837.9 | 100.6 | 121 | | 625.1 | 1256.7 |
| 2022 | 3143.8 | 184.7 | 857.8 | 56.4 | 202 | | 510.3 | 271.3 |
| 2023 (prediction) | 3087.8 | 184.8 | 782.9 | 50.5 | 105.1 | | 458.8 | 524.4 |

Source: based on data from [17].

Expenditures on the conservation of the nature reserves from the state budget in 2015–2020 fluctuated in the range of 1.3–8.3%, increasing to 10.2% (+1.9%) in 2021 and to 18.2% (+8.0%) in 2022. These expenditures, at the expense of local budgets, are characterized by a gradual decrease from 4.0% in 2015 to 1.3% in 2018, and an increase from 1.3% to 11.0% (+9.7%) during 2018–2022. Also, during this period, the expenditures on the conservation of nature reserves increased by 104.2% in the state budget; by 44.2% in the local budget, and by 99.1% in the consolidated budget. In 2023, the above expenditures are expected to decrease.

Fundamental and practical research and development in the field of environmental protection is financed exclusively from the state budget and accounted for less than 5.0% in the period 2015–2022. The existing funding system should be reviewed, as research and development expenditures should also be financed from the local budget, taking into account the territorial specifics of practical testing of scientific research.

The expenditures on other environmental protection activities are largely financed from local budgets and accounted for 30.7–52.2% of the total expenditures in this area in 2015–2022. In the structure of state budget expenditures, their share decreased by 5.6% in 2016, increased by 4.2% in 2017 and by 6.6% in 2018 compared to the previous year, decreased to 7.6% by 2021, and increased to 10.8% in 2022. According to the consolidated budget, these expenditures increased by 7.0% in 2018–2019. This was due to a 24.9% increase in local expenditures and an 18.2% decrease in state expenditures. In 2020, the expenditures decreased by 33.2% in the consolidated budget, and in 2021, they increased by 6.5% compared to the previous period. In 2022, there was a 58.5% reduction in expenditures in general, including 18.4% from the state budget and 78.4% from the local budget. The forecast for 2023 shows an increase in expenditures by 25.8% overall.

An analysis of capital investment in environmental protection shows that in 2016, compared to 2015, it increased by 74.5%; in 2017–2018, it decreased by 17.7% and 8.6%, respectively; in 2019, it increased by 61.4%, followed by a decrease of 18.6% in 2020 and an increase of 6.6% in 2021; in 2022, there was a significant decrease of 54.3%, followed by an increase of 28.5% in 2023 (Figure 3).

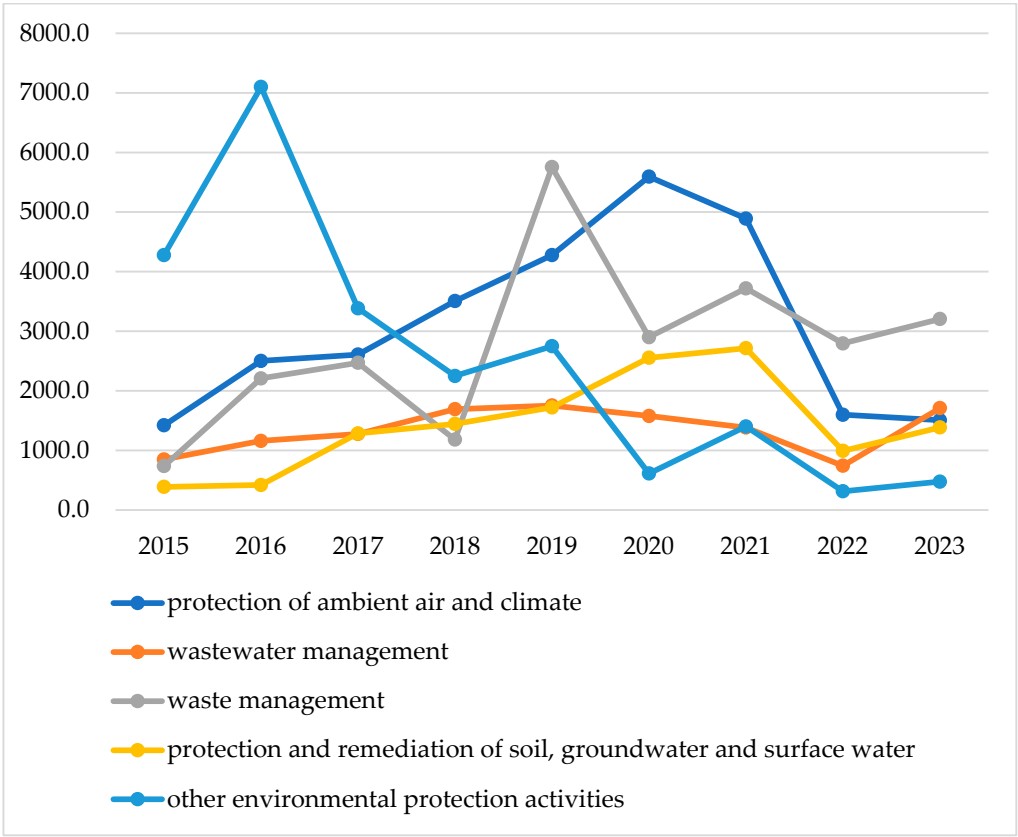

**Figure 3.** Capital investments in environmental protection, millions UAH). Source: based on data from [18].

The structure of investments in 2015–2017 was dominated by other measures, with shares of 55.7%, 53.0%, and 30.7%, respectively; in 2018, 2020–2021—by investments in air protection and climate change (34.8%, 42.3%, and 34.7%, respectively); in 2019, 2022–2023—by investments in waste management (35.4%, 43.4%, and 38.7%, respectively).

To assess the effectiveness of the state's policy in the field of environmental protection, we analyzed the dependence of pollutant emissions on the amount of environmental expenditures, investments in this area, and revenues from environmental tax. The atmospheric air and the costs and revenues associated with it were chosen to be the object of this study (Table 3). The reasonableness of this choice is confirmed by the official statistical data of Ukraine. In addition, it was determined that emissions into the atmosphere can be regulated depending on the activities of business entities and the population as a whole.

**Table 3.** Air pollutant emissions, expenditures and investments in air protection, and environmental tax revenues in Ukraine.

| Year | Emissions of Major Pollutants into the Atmosphere, Thousands of Tons (y) | Expenditures on Air Protection, Millions EUR ($x_1$) | Investments in Air and Climate Protection, Millions EUR ($x_2$) | Revenues from the Environmental Tax Levied on Air Emissions, Millions EUR ($x_3$) |
|---|---|---|---|---|
| 2015 | 4521.3 | 62.7 | 58.7 | 48.9 |
| 2016 | 4686.6 | 62.2 | 88.5 | 110.7 |
| 2017 | 4230.6 | 70.1 | 86.9 | 85.4 |
| 2018 | 4121.2 | 90.2 | 109.1 | 80.3 |
| 2019 | 4108.3 | 102.4 | 147.7 | 124.7 |
| 2020 | 3675.3 | 77.2 | 181.7 | 98.9 |
| 2021 | 4521.3 | 62.7 | 58.7 | 48.9 |

Source: systematized according to [15,18].

First, we tested the existence of a relationship between the dependent and independent variables (Table 4).

**Table 4.** Results of the correlation analysis.

|  | y | $x_1$ | $x_2$ | $x_3$ |
|---|---|---|---|---|
| y | 1 |  |  |  |
| $x_1$ | −0.54587 | 1 |  |  |
| $x_2$ | −0.86366 | 0.597439 | 1 |  |
| $x_3$ | −0.22517 | 0.501104 | 0.620521 | 1 |

Source: calculated using Microsoft Excel.

The Chaddock scale was used to interpret the level of the interdependence. The results show that there is a high correlation between the performance indicator and $x_2$, a medium correlation between y and $x_1$, and a very close correlation between y and $x_3$. It is worth noting the existence of an inverse relationship between the performance attribute and all the factor attributes. For further analysis, we selected $x_2$ as the factor that has the greatest impact on air pollutant emissions (−0.86).

Furthermore, based on the regression analysis, the dependence of air pollutant emissions on investments in this area was determined (Table 5).

**Table 5.** Results of the regression analysis.

| Indicator | Value of $x_2$ |
|---|---|
| Multiple R | 0.863664 |
| R-square | 0.745916 |
| Normalized R-squared | 0.682395 |
| Standard error | 199.6411 |
| Observations | 6 |

Source: calculated using Microsoft Excel.

The multiple correlation coefficient shows the total correlation between y and $x_2$ (0.863664) and indicates a strong relationship between the independent and dependent variables. The coefficient of determination is 0.745916 and shows the overall quality of the model and that the estimated parameters of the model are 74.6% explained by the dependence between the estimated parameters. The rest (25.4%) are inherent in factors not taken into account in the proposed model. The described indicators confirm the regularity of the studied dependence.

The results of the analysis of variance are presented in Table 6.

**Table 6.** Results of the analysis of variance.

|  | df | SS | MS | F | Significance of F |
|---|---|---|---|---|---|
| Regression | 1 | 468,029.2 | 468,029.2 | 11.74283 | 0.026614 |
| Balance | 4 | 159,426.3 | 39,856.58 |  |  |
| Together | 5 | 627,455.5 |  |  |  |

Source: calculated using Microsoft Excel.

The adequacy of the model was confirmed by the Fisher's criterion. When comparing the observed Fisher's criterion with the tabulated one, it was found that, with a reliability coefficient of 0.95 and significance of the hypothesis of 0.05, the calculated value of F (11.74283) is greater than the tabulated value of 0.026614. And since the significance of F is less than 0.05, it can be argued that the model is adequate according to the Fisher criterion, with a reliability level of 0.95.

The existence of a connection within this model is confirmed by the correlation coefficients (Table 7).

**Table 7.** Table of coefficients.

|  | Ratios | Standard Error | t-Statistic | *p*-Value | Bottom 95% | Top 95% |
|---|---|---|---|---|---|---|
| Y-section | 4984.504 | 236.4542 | 21.08021 | $2.99 \times 10^{-5}$ | 4328.002 | 5641.006 |
| $x_1$ | −6.78489 | 1.979959 | −3.42678 | 0.026614 | −12.2821 | −1.28764 |

Source: calculated using Microsoft Excel.

From the table, the linear regression coefficients were determined: $b_0$ = 4984.504; $b_1$ = −6.78489. According to Student's criterion, the coefficient is statistically significant. Based on the linear model, the predicted values in (Table 8) were determined.

**Table 8.** Projected emissions of pollutants into the atmosphere.

| Year | Emissions of Major Pollutants into the Atmosphere, Thousands of Tons (y) | Investments in Air and Climate Protection, Millions EUR ($x_2$) | Emissions of Major Pollutants into the Atmosphere, Thousands of Tons (Predicted Value) |
|---|---|---|---|
| 2015 | 4521.3 | 58.7 | 4586.0 |
| 2016 | 4686.6 | 88.5 | 4384.3 |
| 2017 | 4230.6 | 86.9 | 4394.8 |
| 2018 | 4121.2 | 109.1 | 4244.5 |
| 2019 | 4108.3 | 147.7 | 3982.2 |
| 2020 | 3675.3 | 181.7 | 3751.5 |

Source: calculated using Microsoft Excel according to [15,18].

The distribution of values on the normal distribution graph is narrow, which confirms the accuracy of the model (Figure 4).

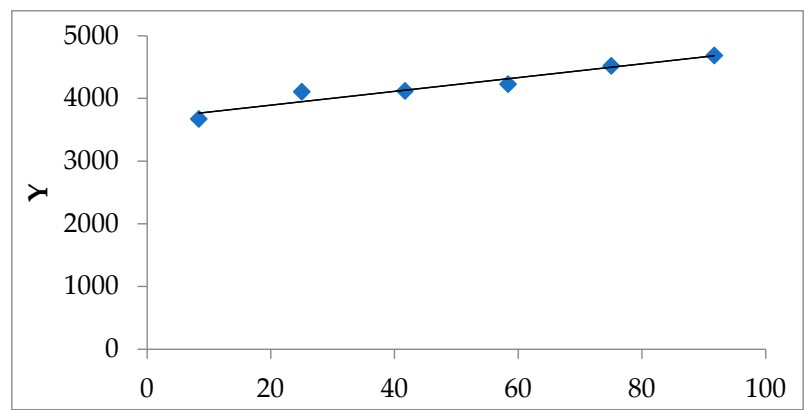

**Figure 4.** Graph of normal distribution. Source: built using Microsoft Excel.

This study found that the emissions of pollutants into the air are inversely proportional to the investments in this area: the higher the amount of investment, the lower the emissions.

As Ukraine aspires to become a member of the European Union, we believed it would be appropriate to conduct a similar analysis of the performance of individual EU member states. Poland and Romania, which are Ukraine's neighbors and cooperate with our country, were chosen for this stage of the study. In addition, these countries are characterized by different levels of economic and social development, which allows us to assess environmental policy in the presence of disproportionate environmental opportunities.

The data for the analysis of Poland's indicators are shown in Table 9.

At the first stage, as in the analysis of Ukraine's indicators, we checked the interconnection between the dependent and independent variables (Table 10).

**Table 9.** Air pollutant emissions, expenditures, investments in air protection, and revenues from pollution taxes in Poland.

| Year | Emissions of Major Pollutants into the Atmosphere, Thousands of Tons (y) | Expenditures on Environmental Protection, Millions EUR ($x_1$) | Investments in Air Protection and Climate Protection, Millions EUR ($x_2$) | Revenues from Pollution Taxes, Millions EUR ($x_3$) |
|---|---|---|---|---|
| 2015 | 340,984.2 | 1629.2 | 112.2 | 725.13 |
| 2016 | 352,343.1 | 1306.5 | 51.5 | 594.98 |
| 2017 | 368,592.5 | 1350.9 | 50.9 | 559.08 |
| 2018 | 367,225.1 | 1645.1 | 145.1 | 538.54 |
| 2019 | 350,689.0 | 1652.0 | 207.1 | 572.65 |
| 2020 | 334,791.2 | 1262.1 | 167.7 | 663.08 |

Source: systematized according to [19].

**Table 10.** Results of the correlation analysis.

|  | y | $x_1$ | $x_2$ | $x_3$ |
|---|---|---|---|---|
| y | 1 | | | |
| $x_1$ | 0.174418 | 1 | | |
| $x_2$ | −0.38216 | 0.483731 | 1 | |
| $x_3$ | −0.83934 | −0.05232 | 0.051072 | 1 |

Source: calculated using Microsoft Excel.

Based on the Chaddock scale, the following was found. There is a weak relationship between the performance attribute and factor $x_1$; between y and $x_2$—a moderate inverse relationship; between y and $x_3$—a high inverse relationship. Further analysis was based on the indicators of the factor $x_3$.

The next step was to conduct a regression analysis (Table 11).

**Table 11.** Results of the regression analysis.

| Indicator | Value of $x_3$ |
|---|---|
| Multiple R | 0.83934206 |
| R-square | 0.704495095 |
| Normalized R-squared | 0.630618868 |
| Standard error | 8267.839883 |
| Observations | 6 |

Source: calculated using Microsoft Excel.

The multiple correlation coefficient proves the existence of a strong relationship between the outcome and factor attributes. The coefficient of determination (R-square) shows that the model parameters are 70.4% explained by the dependence of y and x3. The determined indicators confirm the regularity of the studied dependence.

The results of the analysis of variance are shown in Table 12.

**Table 12.** Results of the analysis of variance.

|  | df | SS | MS | F | Significance of F |
|---|---|---|---|---|---|
| Regression | 1 | 651,864,581.6 | 651,864,581.6 | 9.53615431 | 0.036643091 |
| Balance | 4 | 273,428,705.3 | 68,357,176.33 | | |
| Together | 5 | 925,293,286.9 | | | |

Source: calculated using Microsoft Excel.

The analysis of Fisher's criterion shows significance of F = 0.036643091 < 0.05, which confirms the adequacy of the model. It should also be noted that, according to this criterion, the calculated value with a model reliability level of 95% is higher than the tabulated value.

The correlation coefficients are presented in Table 13.

**Table 13.** Table of coefficients.

| | Ratios | Standard Error | t-Statistic | *p*-Value | Bottom 95% | Top 95% |
|---|---|---|---|---|---|---|
| Y-section | 450,013.033 | 31,777.3782 | 14.1614274 | 0.00014435 | 361,784.89 | 538,241.179 |
| $x_1$ | −160.2462 | 51.89208442 | −3.08806644 | 0.03664309 | −304.3217 | −16.170680 |

Source: calculated using Microsoft Excel.

The following linear regression coefficients were obtained: $b_0$ = 450,013.033; $b_1$ = −160.2462. The test by Student's criterion showed the statistical significance of $b_0$ and $b_1$. The calculated values of the pollutant emissions are shown in Table 14.

**Table 14.** Projected emissions of pollutants into the atmosphere.

| Year | Emissions of Major Pollutants into the Atmosphere, Thousands of Tons (y) | Revenues from Pollution Taxes, Millions EUR ($x_3$) | Emissions of Major Pollutants into the Atmosphere, Thousands of Tons (Predicted Value) |
|---|---|---|---|
| 2015 | 340,984.2 | 725.13 | 333,813.7 |
| 2016 | 352,343.1 | 594.98 | 354,669.7 |
| 2017 | 368,592.5 | 559.08 | 360,422.6 |
| 2018 | 367,225.1 | 538.54 | 363,714.0 |
| 2019 | 350,689.0 | 572.65 | 358,248.0 |
| 2020 | 334,791.2 | 663.08 | 343,757.0 |

Source: calculated using Microsoft Excel according to [19].

The accuracy of the model is confirmed by a normal distribution graph with a narrow distribution of values (Figure 5).

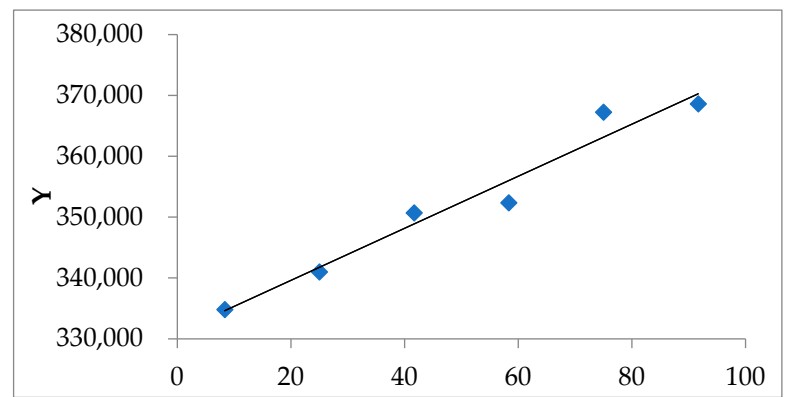

**Figure 5.** Graph of the normal distribution. Source: built using Microsoft Excel.

Romania was chosen as another country for comparative analysis. The initial information for the calculations is presented in Table 15.

**Table 15.** Air pollutant emissions, expenditures, investments in air protection, and revenues from pollution taxes in Romania.

| Year | Emissions of Major Pollutants into the Atmosphere, Thousands of Tons (y) | Expenditures on Environmental Protection, Millions EUR ($x_1$) | Investments in Air Protection and Climate Protection, Millions EUR ($x_2$) | Revenues from Pollution Taxes, Millions EUR ($x_3$) |
|---|---|---|---|---|
| 2015 | 104,264.8 | 1111.2 | 104.6 | 8.21 |
| 2016 | 100,113.4 | 1261.2 | 18.3 | 10.47 |
| 2017 | 102,233.3 | 1220.4 | 6.9 | 9.78 |
| 2018 | 102,372.8 | 1255.9 | 4.7 | 9.45 |
| 2019 | 98,747.1 | 1688.3 | 4.9 | 8.85 |
| 2020 | 94,137.7 | 1823.2 | 6.2 | 9.14 |

Source: systematized according to [19].

The results of the correlation analysis for Romania are shown in Table 16.

**Table 16.** Results of the correlation analysis.

|  | y | $x_1$ | $x_2$ | $x_3$ |
|---|---|---|---|---|
| y | 1 |  |  |  |
| $x_1$ | −0.931071536 | 1 |  |  |
| $x_2$ | 0.544238461 | −0.517229923 | 1 |  |
| $x_3$ | −0.115870636 | −0.135166097 | −0.609250925 | 1 |

Source: calculated using Microsoft Excel.

According to the Chaddock scale, there is a close relationship with factor $x_1$, a significant relationship with factor $x_2$, and a weak relationship with factor $x_3$. Also, the results of the correlation analysis show a direct relationship with factor $x_2$ and an inverse relationship with factors $x_1$ and $x_3$. For the regression analysis, factor $x_1$ was chosen, which has the greatest impact on y. The results are presented in Table 17.

**Table 17.** Results of the regression analysis.

| Indicator | Value of $x_1$ |
|---|---|
| Multiple R | 0.931071536 |
| R-square | 0.866894204 |
| Normalized R-squared | 0.833617755 |
| Standard error | 1461.320607 |
| Observations | 6 |

Source: calculated using Microsoft Excel.

The total correlation between y and $x_1$ according to the correlation coefficient is 0.931071536, which confirms the existence of a strong relationship between the variables. According to the R-squared data, it was determined that the estimated model parameters are 86.7% explained by the outcome and factor attributes; 13.3% are accounted for by other factors.

The next step was to conduct an analysis of variance (Table 18).

**Table 18.** Results of the analysis of variance.

|  | df | SS | MS | F | Significance of F |
|---|---|---|---|---|---|
| Regression | 1 | 55,631,419.56 | 55,631,419.56 | 26.0512835 | 0.006962956 |
| Balance | 4 | 8,541,831.664 | 2,135,457.916 |  |  |
| Together | 5 | 64,173,251.23 |  |  |  |

Source: calculated using Microsoft Excel.

A comparison of the calculated and tabulated values by Fisher's criterion showed that the model is adequate, since 26.0512835 > 0.006962956 and 0.006962956 < 0.05, with a reliability coefficient of 95% and a hypothesis significance of 5%.

The calculated correlation coefficients are shown in Table 19.

**Table 19.** Table of coefficients.

|  | Ratios | Standard Error | t-Statistic | *p*-Value | Bottom 95% | Top 95% |
|---|---|---|---|---|---|---|
| Y-section | 116,392.9 | 3206.693 | 36.29686 | $3.44 \times 10^{-6}$ | 107,489.7 | 125,296.1 |
| $x_1$ | −11.5414 | 2.261221 | −5.10405 | 0.006963 | −17.8195 | −5.26322 |

Source: calculated using Microsoft Excel.

The following linear regression coefficients were obtained: $b_0$ = 116,392.9; $b_1$ = −11.5414. Student's criterion confirmed the statistical significance of the coefficients.

Based on the linear model, the predicted values in Table 20 were determined.

**Table 20.** Projected emissions of pollutants into the atmosphere.

| Year | Emissions of Major Pollutants into the Atmosphere, Thousands of Tons (y) | Expenditures on Environmental Protection, Millions EUR ($x_3$) | Emissions of Major Pollutants into the Atmosphere, Thousands of Tons (Predicted Value) |
|---|---|---|---|
| 2015 | 104,264.8 | 1111.2 | 103,568.1 |
| 2016 | 100,113.4 | 1261.2 | 101,836.9 |
| 2017 | 102,233.3 | 1220.4 | 102,307.8 |
| 2018 | 102,372.8 | 1255.9 | 101,898.1 |
| 2019 | 98,747.1 | 1688.3 | 96,907.6 |
| 2020 | 94,137.7 | 1823.2 | 95,350.6 |

Source: calculated using Microsoft Excel according to [19].

The narrow distribution of values on the normal distribution graph proves the accuracy of the model (Figure 6).

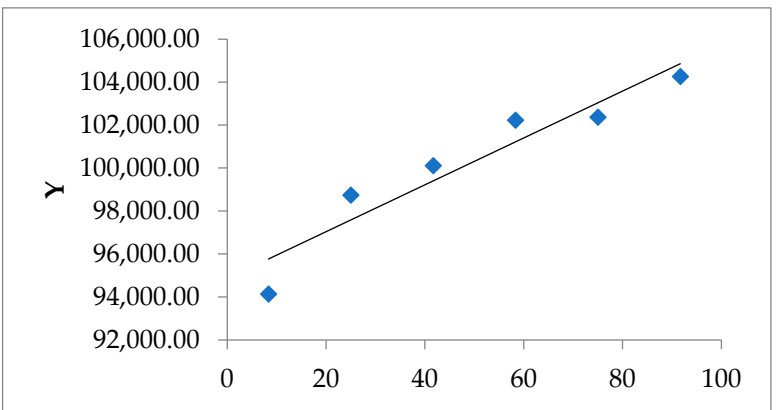

**Figure 6.** Graph of normal distribution. Source: built using Microsoft Excel.

## 4. Discussion

We agree with the position of Babichenko V., Glukhova V., and Kravchenko K., who noted that to ensure sustainable development, it is necessary to maintain a rational balance between the resources used by humanity and the problems that arise in the process of their use. Therefore, it is important to find ways to compensate for the damage caused. The need for greater efficiency of environmental measures and increased sources for their financing indicate the insufficient stability of the environment [3].

Scholars such as Bredikhina V. [20], Vitovska I. [21], and Poyasnik G. [22] have emphasized the importance of public administration in the field of environmental policy regulation. The use of various economic and legal instruments in the field of regulation of environmental relations is inextricably linked to environmental management, the success and effectiveness of which is determined by the effectiveness of measures to protect the environment and preserve natural resources and ecosystems [20]. In modern conditions, it is important to achieve effective coherence and the cooperation of all institutions responsible for the management and coordination of measures for the protection and rational use of natural resources of Ukraine [21]. The role of public environmental management is crucial in addressing the complex problems caused by human activity and climate change. However, global instability complicates the effective conduct of environmental management [22].

An indisputable argument was made by Yashkina V. [23], emphasizing that understanding the various sources and instruments of financing for adaptational measures based on an ecosystemic and nature-oriented approach will contribute not only to the creation of an effective and balanced portfolio of climate finance for climate change adaptation but also the development of a new method of forming a national budget aimed at mitigating and adapting to climate change [23].

We fully agree with the position of Babichenko V. and Glukhova V., who argued that the issue of financial support for environmental protection in modern conditions is defined as one of the most important, since it should not only cover the damage caused to the environment but also contribute to the restoration of natural resources and ensure the sustainable environmentally friendly development of society [3].

It is worth noting that, currently, the funding of environmental protection expenditures is very insufficient. Yaroshevych N. and Yakymiv A. have taken a similar position, stating that the financing of environmental protection from the budget is insufficient and needs to be increased [9]. Constant changes in the structure and powers of the central executive body responsible for the formation and implementation of state policy in the field of environmental protection, as well as significant fluctuations in the structure of state budget expenditures on environmental protection and in the structure of financing of targeted budget "environmental" programs indicate inconsistency in the priorities of the state's environmental policy.

Kovshun N. and Pyatki N. also argued that in Ukraine, the focus on budgetary resources for the financial support of environmental protection measures is unpromising. Additionally, enterprises implement environmental protection measures only if it is economically beneficial for them. At the same time, in order to fulfil Ukraine's international obligations in the field of environmental protection, companies that need to bring their operations to high European standards, which requires significant expenditures, expect the state to help them. This is why it is advisable to develop appropriate means of economic incentives and their legislative consolidation, which would make it possible to solve the problem of financial support for environmental protection by diversifying the sources of funding [24].

## 5. Conclusions

The main challenges in the field of environmental finance are insufficient funding, limited financial resources, and changes in the ratio of environmental tax distribution, which lead to a low efficiency of using funds for environmental protection measures [5].

The ability of Ukraine to provide financial support for the implementation of the environmental management strategy will largely depend on what steps can be taken within the existing organizational and legal structure for financing environmental activities. Therefore, understanding this structure is an important step in developing the necessary economic mechanisms to support and implement a strategy for the rational use of natural resources [25].

Ensuring a balanced ecological and economic development at different levels of economic activity is possible by achieving economic efficiency in financing environmental protection measures. In particular, the introduction of a methodology for assessing the effectiveness of air purification measures at the level of enterprise will allow for evaluating and balancing costs and benefits [11].

The results of the correlation and regression analyses show that Ukraine's financial and environmental policies are not effective enough. This is due to the irrational interrelation of pollutant emissions and the funds invested in this area and the solution of these problems. In this study, it was found that the obtained models are adequate and can be used to build future predictions of pollutant emissions.

Based on the analysis of the current financial and environmental policies, taking into account the situation in Ukraine after the beginning of the full-scale invasion of the Russian Federation, the trends in the EU countries, the study of the legislative framework, regulations, and the works of scholars and practitioners, we propose the following directions for the development of financial and environmental policies:

- The post-war restoration of the environmental situation should be carried out on the basis of sustainable development, focusing on the European Green Deal;
- The triple objective should be implemented: environmental restoration, minimization of negative climate change, and balanced use of resources;

- The powers of the relevant ministry should be expanded, with a focus on the strategic goals of state policy in this area in order to strengthen cooperation with international institutions in solving environmental problems;
- Methodological recommendations should be developed in accordance with international standards for assessing the real state of the environment coupled with its financial interpretation;
- A fundamental reform of the system of allocating funds for environmental purposes should be carried out by means of identifying specific priority areas and setting clear restrictions on the direction of funds;
- State and local budgets should provide funds for environmental protection measures based on the real needs of each individual region.

Implementation of the proposed measures will allow Ukraine to accelerate its progress toward achieving the Sustainable Development Goals, address current environmental issues, and develop strategies for at least 3–5 years. Taking these steps will also allow Ukraine to follow a common course with the EU and, accordingly, develop environment policies, and find ways to use resources in a balanced way.

**Author Contributions:** Conceptualization, O.L.; Methodology, A.S.; Formal analysis, V.L.; Data curation, V.L.; Writing—original draft, O.L. All authors have read and agreed to the published version of the manuscript.

**Funding:** This research received no external funding.

**Institutional Review Board Statement:** Not applicable.

**Informed Consent Statement:** Not applicable.

**Data Availability Statement:** Data are contained within the article.

**Acknowledgments:** The authors would like to thank all the participants in the research activities.

**Conflicts of Interest:** The authors declare no conflicts of interest.

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
