# Peer review of "The Efficiency of Financing Environmental Protection Measures in the Context of Ukraine’s Future Membership in the EU"

_sustainability, doi:10.3390/su16146090_

Round 1

Reviewer 1 Report

Comments and Suggestions for Authors

The paper addresses crucial points and timely issues that are at the forefront of academic discussion, potentially making a significant impact on the academic community. However, some necessary changes remain.

1. The abstract should include the motivation for the concern, the critical control variables, the method of analysis, the data used, the findings, and the policy implications—recommendations for further optimization of this section and eliminating duplication (e.g., abstract).

2. The paper's first part (Introduction) section is concise and needs improvement. The introduction must clearly state the paper's purpose, significance, contribution, and framework in a separate paragraph. Firstly, considering the choice of sample study, why was Ukraine chosen as the sample for the analysis instead of other developed and developing countries? It is recommended that a more detailed explanation be provided. Secondly, it is recommended that the content of the literature review be provided. Using more recent literature would make the work more reader-friendly. Thirdly, it is suggested that the study's purpose, significance, and contribution be added. Fourth, the last paragraph of the introduction should provide an outline of the paper.

3. The second part of the article (Materials and Methods) suggests adding an empirical model to clarify the empirical framework of the literature. In addition, the authors should provide a clear theoretical foundation before presenting the empirical model.

4. The third part of the article (Results) suggests adjustments and additions. In Figure 1, a part of the illustration ("an environmental tax levied on carbon dioxide emissions from stationary sources of pollution") is missing for Ukraine's environmental tax revenues. Secondly, the text at the top of Table 2 indicates that the data in Table 2 are "the consolidated budget figures for the general and special funds." Still, the data in Table 2 are "State, Local, Consolidated." It is recommended that this be clarified. Thirdly, in the third paragraph under Table 1, "Expenditures as a percentage of GDP," why is the percentage of GDP used? Fourth, in the third paragraph under Table 2, "Fundamental and practical research and developments in the field," only the state budget is mentioned, and it is suggested that more relevant explanations be added. Fifth, Figure 1, Table 1, and Table 2 provide detailed descriptions of the data for the core variables (expenditures on environmental protection, revenues from environmental taxes), and it is recommended that investment-related data be added. Sixth, the heading of Table 10 contains redundancies that need to be deleted. Seventh, in Table 14, there is an additional punctuation mark at the end of the upper paragraph, and it is recommended to review the use of punctuation in the whole text.

5. The discussion, research results, and policy implications should be improved.

6. Recheck the paper for grammatical, spelling, or formatting errors.

Comments on the Quality of English Language

Minor editing of English language required

Author Response

Thank you very much for your time spent on our research, for all the recommendations and suggestions you have provided. The opinion of highly qualified experts is very important to us. We tried to take all the recommendations into account as much as possible when making the changes to the article. Comments 1: The abstract should include the motivation for the concern, the critical control variables, the method of analysis, the data used, the findings, and the policy implications —recommendations for further optimization of this section and eliminating duplication (e.g., abstract). Response 1: Thank you for your comments. We have tried to take them into account as much as possible, in particular, the abstract has been supplemented with research methods, background information and recommendations for improving financial and environmental policy (page 1). Comments 2: The paper's first part (Introduction) section is concise and needs improvement. The introduction must clearly state the paper's purpose, significance, contribution, and framework in a separate paragraph. Firstly, considering the choice of sample study, why was Ukraine chosen as the sample for the analysis instead of other developed and developing countries? It is recommended that a more detailed explanation be provided. Secondly, it is recommended that the content of the literature review be provided. Using more recent literature would make the work more reader-friendly. Thirdly, it is suggested that the study's purpose, significance, and contribution be added. Fourth, the last paragraph of the introduction should provide an outline of the paper. Response 2: Thank you for these comments, we agree with them and have taken them into account in the paper, namely: we have justified the choice of Ukraine as the object of research; we have indicated the scholars who have studied this topic, the purpose and contribution of the study; we have added an outline of the article. The addendum can be found on page 2 (paragraphs 2, 4, 5).

Comments 3: The second part of the article (Materials and Methods) suggests adding an empirical model to clarify the empirical framework of the literature. In addition, the authors should provide a clear theoretical foundation before presenting the empirical model.

Response 3: Thank you for your comments. We have taken them into account in the paper as follows: we have added empirical models, having previously justified the essence of correlation and regression models. The changes are presented on pages 3 (paragraphs 5, 6) and 4 (paragraphs 1, 2).

Comments 4: The third part of the article (Results) suggests adjustments and additions. In Figure 1, a part of the illustration ("an environmental tax levied on carbon dioxide emissions from stationary sources of pollution") is missing for Ukraine's environmental tax revenues. Secondly, the text at the top of Table 2 indicates that the data in Table 2 are "the consolidated budget figures for the general and special funds." Still, the data in Table 2 are "State, Local, Consolidated." It is recommended that this be clarified. Thirdly, in the third paragraph under Table 1, "Expenditures as a percentage of GDP," why is the percentage of GDP used? Fourth, in the third paragraph under Table 2, "Fundamental and practical research and developments in the field," only the state budget is mentioned, and it is suggested that more relevant explanations be added. Fifth, Figure 1, Table 1, and Table 2 provide detailed descriptions of the data for the core variables (expenditures on environmental protection, revenues from environmental taxes), and it is recommended that investment-related data be added. Sixth, the heading of Table 10 contains redundancies that need to be deleted. Seventh, in Table 14, there is an additional punctuation mark at the end of the upper paragraph, and it is recommended to review the use of punctuation in the whole text.

Response 4: We fully agree with the comments provided and have tried to take them into account as much as possible in the article. Part of the illustration in Figure 1 has been added (page 5). The text at the top of Table 2 "consolidated budget indicators by general and special funds" has been deleted (page 7). In the third paragraph of Table 1, "Expenditures as a percentage of GDP", GDP is indicated, as budget deficits are usually measured as a percentage of GDP. Therefore, the definition of this ratio is relevant from the point of view of analysing the real change in the value of expenditures. In the third paragraph of Table 2 "Fundamental and Practical Research and Development in the Sector", an addition was made to the expediency of ensuring that these costs can be financed from local budgets (pages 8-9). The article is supplemented with information on the structure of capital investments in environmental protection (Figure 2, pages 9-10). The heading of Table 10 has been removed from redundant information (page 13). In Table 14, an additional punctuation mark was removed at the end of the top paragraph (page 14).

Comments 5: The discussion, research results, and policy implications should be improved.

Response 5: Thank you for your comments. We have made additions to the discussion and results of the study.

Comments 6: Recheck the paper for grammatical, spelling, or formatting errors.

Response 6: Thank you for your comments. We have checked the article and corrected the errors.  

Reviewer 2 Report

Comments and Suggestions for Authors

The paper entitles "Efficiency of financing environmental protection measures in the context of Ukraine's future membership in the EU". I worry about the new contribution in this study. Therefore, I have to reject this paper. The reasons are as follows:

1. The research depth of the paper is not sufficient, as it only simply analyzes the efficiency of financing environmental protection measures from three aspects: environmental expenditure, investment in this field, and environmental taxation.

2. There are some obvious errors in the paper. For example, in line 293, "There is a weak inverse relationship between the performance attribute and factor x1;......" should be "There is a weak relationship between the performance attribute and factor x1;......".

3. The discussion section does not provide an in-depth analysis of the research findings of the paper.

Comments on the Quality of English Language

Moderate editing of English language required.

Author Response

Dear Reviewer,

Thank you very much for your time spent on our research, for all the recommendations and suggestions you have provided. The opinion of highly qualified experts is very important to us. We tried to take all the recommendations into account as much as possible when making the changes to the article. Comments 1: The research depth of the paper is not sufficient, as it only simply analyzes the efficiency of financing environmental protection measures from three aspects: environmental expenditure, investment in this field, and environmental taxation.

Response 1: Thank you for your comments. We would like to clarify some of the content of the article. The paper presents an analysis of the structure of environmental expenditures by budget and functional classification. The article is also supplemented by an analysis of capital investments in environmental protection (pages 9-10).

A separate section examines the effectiveness of Ukraine's environmental policy by means of a correlation and regression analysis of the dependence of pollutant emissions on environmental expenditures, investments in this area, and environmental tax revenues. Since Ukraine is seeking to become a member of the EU, to compare the effectiveness of its environmental policy, we also studied the indicators of Poland and Romania, which are Ukraine's neighbours and cooperate with our country. In addition, these countries are characterised by different levels of economic and social development, which allows us to assess environmental policy in the presence of disproportionate environmental opportunities. Comments 2: There are some obvious errors in the paper. For example, in line 293, "There is a weak inverse relationship between the performance attribute and factor x1;......" should be "There is a weak relationship between the performance attribute and factor x1;......". Response 2: Thank you for your comments. We tried to correct the errors made in the work as much as possible, including the error in line 293. Comments 3: The discussion section does not provide an in-depth analysis of the research findings of the paper. Response 3: Thank you for your comments. The discussion section is supplemented by an analysis of the research results and the views of scholars on the topic (page 18).  

Round 2

Reviewer 1 Report

Comments and Suggestions for Authors

I'm very sorry; I couldn't see any apparent traces of direct revisions from this draft. Based solely on the author's feedback, judging every detail is difficult. This revised draft could have effectively addressed many of the significant concerns raised in the previous draft.

I have carefully studied this paper again and still found many significant concerns that must be revised. I hope to receive sufficient attention from the author.

Could the author provide a more detailed explanation for why the results for the same region in Model 3 and Model 4 on page 4 are exactly the same? This would help to clarify what seems to be an unreasonable outcome.

The tables of the paper, such as Table 2, need to be further improved to make them more readable.

The paper's conclusion is suggested to be based on correlation testing, which leads to unreliable conclusions. This is a significant gap from published papers in this field.

Comments on the Quality of English Language

I'm very sorry; I couldn't see any apparent traces of direct revisions from this draft. Based solely on the author's feedback, judging every detail is difficult. This revised draft could have effectively addressed many of the significant concerns raised in the previous draft.

I have carefully studied this paper again and still found many significant concerns that must be revised. I hope to receive sufficient attention from the author.

Could the author provide a more detailed explanation for why the results for the same region in Model 3 and Model 4 on page 4 are exactly the same? This would help to clarify what seems to be an unreasonable outcome.

The tables of the paper, such as Table 2, need to be further improved to make them more readable.

The paper's conclusion is suggested to be based on correlation testing, which leads to unreliable conclusions. This is a significant gap from published papers in this field.

Author Response

Comments 1: Could the author provide a more detailed explanation for why the results for the same region in Model 3 and Model 4 on page 4 are exactly the same? This would help to clarify what seems to be an unreasonable outcome. Response 1 : Thank you very much for your comment, unfortunately there was a technical error that has been corrected (page 4). Model 4 refers to Romania. Comments 2: The tables of the paper, such as Table 2, need to be further improved to make them more readable. Response 2: Thank you for your comment. Table 2 has been changed. Some information is presented in the form of Figure 2 (pages 7-8).

Comments 3: The paper's conclusion is suggested to be based on correlation testing, which leads to unreliable conclusions. This is a significant gap from published papers in this field.

Response 3: Thank you for your comment! The conclusions have been amended (pages 18-19). We would also like to point out that the conclusions are based not only on the data from the correlation and regression analysis, but also on the assessment of the current financial and environmental policy, the study of the regulatory framework and the study of the works of scientists and practitioners in this area. 

Reviewer 2 Report

Comments and Suggestions for Authors

The author has carefully revised and improved the paper. My review decision is to accept it.

Comments on the Quality of English Language

Minor editing of English language is required.

Author Response

Dear reviewer,

thank you!

Round 3

Reviewer 1 Report

Comments and Suggestions for Authors

This revised draft incorporates the comments from the previous round and makes significant revisions, resulting in substantial improvements. Thank you to the author for their careful revision. The major issues mentioned in the last draft have been adequately addressed and responded to. I have no new significant concerns regarding this draft.